# TLR5 participates in the TLR4 receptor complex and promotes MyD88-dependent signaling in environmental lung injury

Salik Hussain[1,2†], Collin G Johnson[1,3†], Joseph Sciurba[1,4†], Xianglin Meng[1,5], Vandy P Stober[1], Caini Liu[6], Jaime M Cyphert-Daly[1,7], Katarzyna Bulek[6,8], Wen Qian[6], Alma Solis[1], Yosuke Sakamachi[1], Carol S Trempus[1], Jim J Aloor[1,9], Kym M Gowdy[1,9], W Michael Foster[7], John W Hollingsworth[7], Robert M Tighe[7], Xiaoxia Li[6], Michael B Fessler[1], Stavros Garantziotis[1]*

[1]National Institute of Environmental Health Sciences, Research Triangle Park, United States; [2]Department of Physiology and Pharmacology, School of Medicine, West Virginia University, Morgantown, United States; [3]Center for Cell and Gene Therapy, Baylor College of Medicine, Houston, United States; [4]Department of Veterinary Medicine, North Carolina State University, Raleigh, United States; [5]Department of ICU, First Affiliated Hospital of Harbin Medical University, Harbin, China; [6]Lerner Research Institute, Cleveland Clinic Foundation, Cleveland, United States; [7]Duke University Medical Center, Durham, United States; [8]Department of Immunology, Faculty of Biochemistry, Biophysics and Biotechnology, Jagiellonian University, Krakow, Poland; [9]East Carolina University Brody School of Medicine, Greenville, United States

*For correspondence:
garantziotis@niehs.nih.gov

†These authors contributed equally to this work

Competing interests: The authors declare that no competing interests exist.

**Abstract** Lung disease causes significant morbidity and mortality, and is exacerbated by environmental injury, for example through lipopolysaccharide (LPS) or ozone ($O_3$). Toll-like receptors (TLRs) orchestrate immune responses to injury by recognizing pathogen- or danger-associated molecular patterns. TLR4, the prototypic receptor for LPS, also mediates inflammation after $O_3$, triggered by endogenous hyaluronan. Regulation of TLR4 signaling is incompletely understood. TLR5, the flagellin receptor, is expressed in alveolar macrophages, and regulates immune responses to environmental injury. Using in vivo animal models of TLR4-mediated inflammations (LPS, $O_3$, hyaluronan), we show that TLR5 impacts the in vivo response to LPS, hyaluronan and $O_3$. We demonstrate that immune cells of human carriers of a dominant negative TLR5 allele have decreased inflammatory response to $O_3$ exposure ex vivo and LPS exposure in vitro. Using primary murine macrophages, we find that TLR5 physically associates with TLR4 and biases TLR4 signaling towards the MyD88 pathway. Our results suggest an updated paradigm for TLR4/TLR5 signaling.

## Introduction

Lung disease is a major contributor to morbidity and mortality worldwide. In the US alone, over 15% of the population suffers from lung disease, at an annual cost of 120,000 deaths, and >$50 billion (*Redd, 2002*; *Kochanek et al., 2011*; *Ford et al., 2015*). Environmental lung injury, for example through inhaled lipopolysaccharide (LPS) or elevated ozone ($O_3$) levels, exacerbates lung disease (*Bell et al., 2004*; *Katsouyanni et al., 1995*; *Hubbell et al., 2005*; *Thorne et al., 2005*). For

**eLife digest** Immune cells in the lung help guard against infections. On the surface of these cells are proteins called TLR receptors that recognize dangerous molecules or DNA from disease-causing microbes such as bacteria. When the immune cells detect these invaders, the TLR receptors spring into action and trigger an inflammatory response to destroy the microbes. This inflammation usually helps the lung clear infections. But it can also be harmful and damage the lung, for example when inflammation is caused by non-infectious substances such as pollutants in the atmosphere.

There are several TLR receptors that each recognize a specific molecule. In 2010, researchers showed that the receptor TLR4 is responsible for causing inflammation in the lung after exposure to pollution. Another receptor called TLR5 also helps activate the immune response in the lung. But it was unclear whether this receptor also plays a role in pollution-linked lung damage.

Now, Hussain, Johnson, Sciurba et al. – including one of the researchers involved in the 2010 study – have investigated the role of TLR5 in immune cells from the lungs of humans and mice. The experiments showed that TLR5 works together with TLR4 and helps trigger an inflammatory response to both pollutants and bacteria. Hussain et al. found that people lacking a working TLR5 receptor (which make up 3–10% of the population) are less likely to experience lung inflammation when exposed to pollution or bacterial proteins that activate TLR4.

These findings suggest that people without TLR5 may be protected from pollution-induced lung injury. Further research into the role of TLR5 could help develop genetic tests for identifying people who are more sensitive to damage from pollution. This information could then be used to determine the likelihood of a patient experiencing certain lung diseases.

example, household LPS levels are directly associated with asthma symptoms and asthma-related health care utilization (*Thorne et al., 2005*). Also, sound epidemiological studies suggest that for every 10 parts-per-billion (ppb) increase in $O_3$ levels there is an associated mortality increase of 0.39–0.87% (*Katsouyanni et al., 1995*; *Levy et al., 2005*). It is therefore imperative to understand the mechanisms of environmentally induced lung injury.

Innate immune activation is a major contributor to lung disease pathogenesis and environmentally-induced exacerbations (*Miller and Peden, 2014*; *Garantziotis and Schwartz, 2010*). Toll-like receptors (TLRs) orchestrate the innate immune response to lung injury by recognizing exogenous pathogen- or endogenous danger-associated molecular patterns. TLR4 is the prototypic receptor for LPS (*Kawai and Akira, 2010*), which is found in particulate-matter pollution and house dust (*Thorne et al., 2005*; *Mueller-Anneling et al., 2004*) and is a major contributor to sepsis-induced lung injury (*Andonegui et al., 2003*; *Baumgarten et al., 2006*). TLR4 also mediates inflammation and airway hyperresponsiveness after $O_3$ exposure (*Garantziotis et al., 2010*), triggered by release of the endogenous sugar hyaluronan (*Garantziotis et al., 2009*).

Regulation of TLR4 signaling is still incompletely understood. TLR4 can heterodimerize with other TLR like TLR2 and TLR6 (*Stewart et al., 2010*; *Wang et al., 2014*); in these cases, the partner TLRs serve to expand the TLR4 ligand spectrum. However, until now there has not been evidence of TLR4 interaction with other TLR, that modulates TLR4 signaling in response to its own ligands. TLR4-TLR5 interaction has been reported once (*Mizel et al., 2003*), wherein TLR4 was shown to promote nitric oxide production after flagellin exposure. We therefore hypothesized that the reciprocal interaction may also be true, that is TLR5 participates in TLR4 signaling after environmental lung injury. TLR5, the prototypic receptor for bacterial flagellin, is expressed in alveolar macrophages (*Shikhagaie et al., 2014*), is induced after injury (*Menendez et al., 2011*) and regulates the immune response to injury (*Burdelya et al., 2008*; *Uematsu et al., 2008*; *Wilson et al., 2012*). TLR5 plays an important role in immunity and metabolism and has been implicated in processes as varied as asthma (*Wilson et al., 2012*), antiviral defense (*Zhang et al., 2014*), ischemia-reperfusion injury (*Parapanov et al., 2015*; *Fukuzawa et al., 2011*), radiation–induced injury (*Burdelya et al., 2008*; *Burdelya et al., 2012*), and regulation of gut immunity (*Uematsu et al., 2008*). Furthermore, known functional genetic polymorphisms in *TLR5* are associated with susceptibility to infections (*Hawn et al., 2003*; *Grube et al., 2013*; *West et al., 2013*) and autoimmune disease

(*Gewirtz et al., 2006*). These findings suggest a clinically relevant role of TLR5 in human immune regulation in the response to injury.

We show that TLR5 deficiency in mice significantly alters the in vivo response to TLR4 activators LPS, hyaluronan and $O_3$. Mechanistically, we show that after ultrapure LPS exposure, TLR5 co-immunoprecipitates with MyD88, TLR4 and LPS. The presence of TLR5 promotes formation of the Myddosome, that is association of MyD88 and IRAK4, and biases TLR4 signaling towards the MyD88 pathway. Finally, we demonstrate that human carriers of a dominant-negative TLR5 allele have decreased inflammatory response to $O_3$ exposure in vivo and LPS exposure in vitro. Our results thus suggest that TLR5 participates in TLR4 signaling and modulates environmental lung injury in disease-relevant exposures that lead to TLR4 activation.

## Results

### TLR5 promotes TLR4-mediated inflammation and airway hyperresponsiveness in vivo

We first investigated the effect of TLR5 on TLR4 signaling in vivo, by exposing Tlr5-deficient mice or wildtype controls to LPS via intraperitoneal administration. As expected, this led to substantial lung

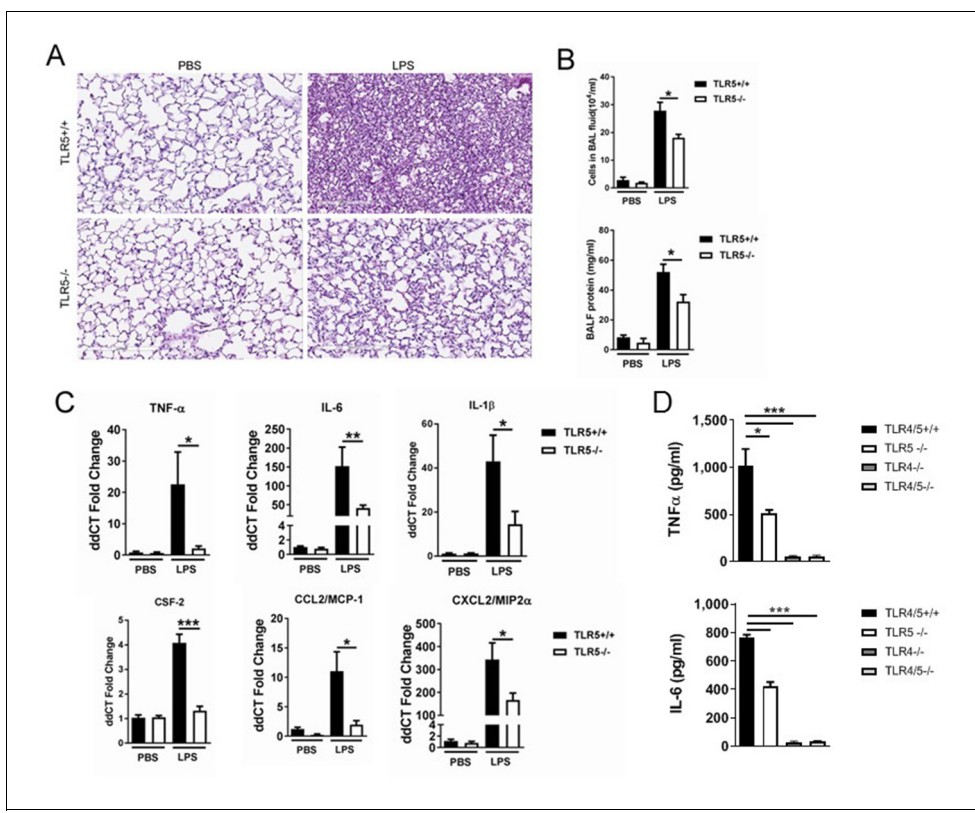

**Figure 1.** TLR5 deficiency ameliorates the inflammatory lung response to systemic LPS at 24 hr after exposure. (A) Hematoxylin-Eosin staining of lung sections demonstrates LPS-induced lung injury is ameliorated in Tlr5-deficient (Tlr5$^{-/-}$) mice. (B) Cellular lung inflammation and lung lavage protein levels are decreased in Tlr5-deficient (Tlr5$^{-/-}$) mice. (C) Real time quantitative PCR analysis of inflammatory cytokines shows a significant decrease in Tlr5-deficient (Tlr5$^{-/-}$) mice. N = 5–8 mice per group, experiment repeated twice. (D) Lung lavage cytokine analysis confirms a significant decrease in inflammatory cytokines after ultrapure LPS exposure in Tlr5-deficient (Tlr5$^{-/-}$) mice. Note absent response in Tlr4$^{-/-}$ and Tlr4/5$^{-/-}$. N = 4–7 mice per group. Data are represented as mean ± s. e.m. and were analyzed by unpaired t test with Welch's correction *p<0.05 ** p<0.01 and ***p<0.001.

The online version of this article includes the following figure supplement(s) for figure 1:

**Figure supplement 1.** TLR5 deficiency ameliorates the inflammatory response to LPS exposure.

**Figure supplement 2.** Immune gene expression profiling dependent on TLR5 after ultrapure LPS exposure.

inflammation in wildtype mice, which was ameliorated in the absence of TLR5 (*Figure 1A*). TLR5 deficiency also ameliorated cellular influx and lung injury as evidenced by lung lavage protein (*Figure 1B*). Furthermore, expression of inflammatory cytokines in the lung was significantly ameliorated in Tlr5-deficient mice (*Figure 1C,D*). This was largely mirrored in a reduction of inflammatory gene expression in the liver (*Figure 1—figure supplement 1A*). To ensure that our results were not affected by obesity-, microbiome- or breeding-related immune perturbations, we performed experiments with mice that were either purchased from a commercial vendor (using C57BL/6 as controls) or bred in our NIEHS colony (using wild-type littermates as controls) and treated some mice with neomycin to reduce bacterial burden in the intestinal tract. Our results did not vary regardless of mouse provenance or antibiotic treatment (*Figure 1—figure supplement 1B*).

To determine if the TLR5 effect on TLR4 signaling has broader biological and clinical relevance in the lung, we explored TLR5-mediated effects on sterile lung injury after exposure to the ambient pollutant, $O_3$. $O_3$ exposure is associated with increased morbidity and mortality in human patients with cardiopulmonary disease (*Katsouyanni et al., 1995*; *Levy et al., 2005*); furthermore, it is now understood that TLR4 mediates the development of inflammation and airway hyperresponsiveness (AHR) after $O_3$ exposure (*Garantziotis et al., 2010*). We used an $O_3$ dose that is equivalent to human exposure during a high-$O_3$ day (*Slade et al., 1997*; *Hatch et al., 1994*). Tlr5-deficient mice had ameliorated airway cytokine expression and almost abolished AHR after $O_3$ exposure (*Figure 2A,B*). Because hyaluronan is the endogenous danger-associated molecular pattern that activates TLR4 and mediates the response after $O_3$ exposure (*Garantziotis et al., 2010*; *Garantziotis et al., 2009*), we then investigated the effect of TLR5 on hyaluronan signaling. Tlr5-deficient mice had substantially reduced inflammatory gene induction and significantly diminished AHR after instilled ultrapure (pharmaceutical grade) hyaluronan exposure (*Figure 3A,B*). We then compared the effects of Tlr4-, Tlr5- and Myd88 deficiency on the hyaluronan response in vitro. We incubated tracheal rings of these strains with short fragments of hyaluronan, which induces hyperresponsiveness to methacholine-induced constriction through a TLR4-MyD88 pathway

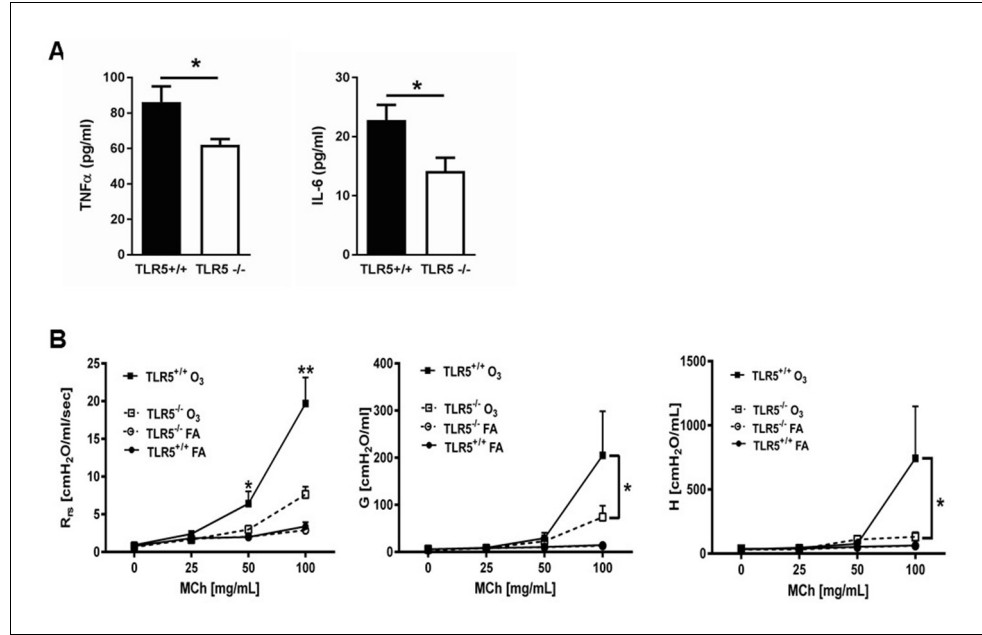

**Figure 2.** TLR5 deficiency ameliorates the in vivo inflammatory response to inhaled $O_3$. (**A**) TNF-α and IL-6 levels in the lung lavage fluid of *Tlr5*-deficient (*Tlr5−/−*) or *Tlr5*-competent (*Tlr5+/+*) mice 24 hr after receiving 3 ppm $O_3$ for 3 hr by inhalation. n = 14 mice for *Tlr5+/+* and n = 12 mice for *Tlr5−/−*, experiment repeated twice. (**B**) Airway physiology measurement (total respiratory resistance $R_{rs}$, tissue damping G and tissue elastance H) to indicated doses of methacholine challenge measured with flexiVent in *Tlr5*-deficient (*Tlr5−/−*) or *Tlr5*-competent (*Tlr5+/+*) mice 24 hr after 2ppm $O_3$ or air (FA) exposure. n = 6 for *Tlr5−/−*FA and *Tlr5−/−*$O_3$ and n = 7 for *Tlr5+/+*FA and *TLlr5+/+*$O_3$, experiment repeated three times. Data are represented as mean ± s.e.m. and were analyzed by unpaired t test with Welch's correction *p<0.05 and **p<0.01 between *Tlr5+/+* and *Tl r5−/−* mice exposed to $O_3$.

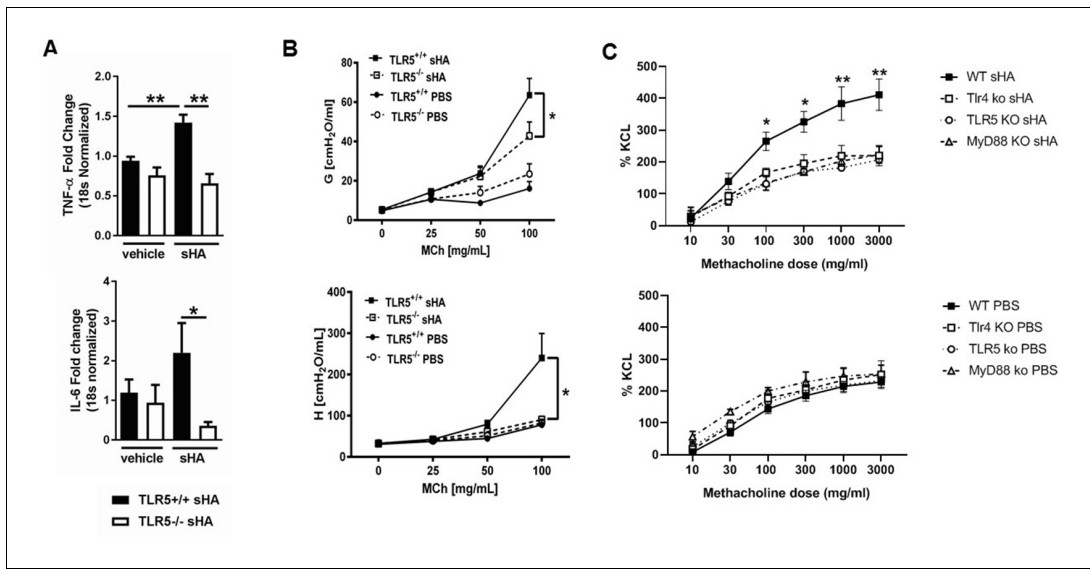

**Figure 3.** TLR5 deficiency ameliorates the in vivo and in vitro inflammatory and airway constrictive response to ultrapure short-fragment hyaluronan (sHA). (**A**) Real time quantitative PCR gene expression of TNF-α and IL-6 in the lung tissues of *Tlr5*-deficient (TLR5[−/−]) or *Tlr5*-competent (*Tlr5*[+/+]) mice 6 hr after exposure to vehicle (PBS) or 50 μl of 3 mg/ml short fragment HA. n = 5 for *Tlr5*[−/−]PBS and *Tlr5*[−/−]sHA and n = 6 for *Tlr5*[+/+]PBS and *Tlr5*[+/+]sHA. Experiment repeated once. (**B**) Airway physiology measurement (tissue damping G and tissue elastance H) to indicated doses of methacholine challenge measured with flexiVent 2 hr after exposure to vehicle (PBS) or 2 mg/ml sHA. n = 5 for *Tlr5*[−/−]PBS and *Tlr5*[−/−]sHA and n = 6 for *TlrR5*[+/+]PBS and *Tlr5*[+/+]sHA, experiment repeated twice. (**C**). Constriction measurement of tracheal rings of C57Bl/6J mice (N = 4–6 per condition), incubated in vitro with sHA, which is known to cause hyperresponsiveness to methacholine and then treated with indicated doses of methacholine. Note that deficiency in Tlr4 and Tlr5 abolishes the tracheal ring response to hyaluronan to levels similar with Myd88 deficiency, suggesting that Tlr4 and Tlr5 play non-redundant roles in hyaluronan-Myd88 signaling. Baseline responsiveness to methacholine is not different between these strains (bottom panel C). Data are represented as mean ± s.e.m. and were analyzed by unpaired t test with Welch's correction (panel A) or ANOVA with Tukey's post-hoc analysis (panels B and C) *p<0.05 and *p<0.01 between groups.

(*Garantziotis et al., 2010*; *Li et al., 2011*) We found that deficiency in either of the 3 molecules abolished the response to hyaluronan (*Figure 3C*) suggesting that TLR4 and TLR5 are non-redundant and necessary for the hyaluronan-MyD88 signaling response.

We then performed a more global analysis of the TLR5 effects on TLR4 signaling. We analyzed gene expression patterns using the NanoString platform (www.nanostring.com) and utilizing the Mouse Innate Immunity Panel Codeset (Ns_Mm_Myeloid_v2.0) and focused specifically on 242 genes that were more than 2-fold upregulated after LPS exposure (*Supplementary files 1a and 1b* and *Figure 1—figure supplement 2*) and sorted them according to magnitude of TLR5 effect. We showed that the presence or absence of functional TLR5 is associated with differential regulation of immune genes in this panel. Interestingly, there was a linear correlation between the magnitude of the TLR5 effect and the proportion of genes that are either published or predicted to be downstream of the NFκB pathway (*Figure 1—figure supplement 2C*): among the genes that were 70–90% upregulated in *Tlr5*-sufficient mice compared to *Tlr5*-deficient mice, almost 90% were in the NFκB pathway, while this proportion fell to 50% among the genes that were not different between genotypes. ($R^2$ = 0.89, p=0.0013). This may suggest that TLR5 preferentially impacts gene expression downstream of NFκB activation. Alternatively, because these data are from whole lung tissue, this could reflect decreased recruitment of inflammatory cells, although we are not aware that inflammatory cells preferentially express NFκB pathway genes. In aggregate, the presented results support that TLR5 promotes TLR4 signaling in several models of TLR4 activation through pathogen- or danger-associated molecular patterns (PAMPs or DAMPs, that is LPS or hyaluronan respectively) and promotes TLR4-mediated inflammation and airway hyperresponsiveness in vivo.

## Genetic TLR5 deficiency in humans impacts TLR4 signaling in vitro and ex vivo

We then investigated the effect of TLR5 in human TLR4 signaling. In humans, a dominant-negative *TLR5* single nucleotide polymorphism (SNP) (rs5744168, *TLR5$^{392STOP}$*) (*Hawn et al., 2003*) is found with a prevalence of 8–10% in Caucasians and 3% in African Americans. We hypothesized that carriers of this SNP may have reduced TLR4-mediated inflammation. We used the Environmental Polymorphisms Registry (*Chulada et al., 2011*), a NIEHS-supported cohort, to recruit carriers of this allele, as well as 'wildtype' controls. Others have reported that whole blood from rs5744168 minor-allele carriers does not differ in the response to LPS compared to 'wildtype' (*West et al., 2013*), and we confirmed this finding (*Figure 4—figure supplement 1A*). We believe this happens because whole blood consists of different cell types, which differentially express TLR5, thereby confounding the effect on TLR4 signaling. We then investigated the effect of this *TLR5* SNP on purified, primary monocyte-derived macrophages. Macrophages from rs5744168 minor-allele carriers had a decreased response to flagellin and ultrapure LPS, but not Pam3CSK4 (*Figure 4A* and *Figure 4—figure supplement 1B*), thus confirming that human *TLR5* genetic variation specifically determines the response to LPS. This was not due to altered expression of TLR4 or CD14, which was not changed by the rs5744168 genotype (*Figure 4—figure supplement 1C*).

We then investigated the effect of TLR5 in $O_3$-induced inflammation in healthy human volunteers. We exposed human volunteers to $O_3$, and isolated alveolar macrophages through bronchoscopy 24 hr after exposure, which represents the peak of $O_3$-induced inflammation and symptoms in humans. *TLR5* expression was modestly increased in alveolar macrophages of human volunteers after $O_3$ exposure (*Figure 4B*, p=0.05 by Wilcoxon pairwise signed rank test). There was no association between *TLR5* expression and *TLR4* expression after $O_3$ exposure. We found, that *TNFα* expression by alveolar macrophages after $O_3$ exposure was not increased in any of the *TLR5*-deficient individuals (rs5744168 minor-allele carriers), while it was increased in wildtype-allele carriers (*Figure 4C* and *Figure 4—figure supplement 1D*).

## TLR5 modulates TLR4-dependent signaling

To investigate the mechanistic role for TLR5 in the response to LPS-induced TLR4 activation, we then compared primary bone marrow derived macrophages (BMDM) from *Tlr5*-deficient and -sufficient mice. *Tlr5*-deficient BMDM had significantly decreased expression of TNFα and IL-6 after ultrapure LPS exposure in vitro, by an average of 30–50% (*Figure 5A*). To ensure that the altered response was not due to LPS contaminants despite the ultrapure preparation, we also assayed *Tlr4*-deficient BMDM and saw no response to ultrapure LPS (*Figure 5A*). The effect of *Tlr5*-deficiency was also observed when using other sources of Tlr4 activation, such as Monophosphoryl Lipid A (*Figure 5—figure supplement 1A*). To ascertain that our observation was not due to an off-target effect of genetic *Tlr5* ablation on macrophage biology, we used RAW264.7 cells, a murine macrophage cell line, which are naturally deficient in *Tlr5* (*McDonald et al., 2007*). We transfected these cells with a murine *Tlr5* construct, or empty vector, and noted that the *Tlr5*-transfected cells had significantly higher TNFα expression in response to ultrapure LPS (and the TLR5 ligand flagellin, as expected). The TLR5-dependent effect was specific to LPS exposure, as responses to poly(I:C) (TLR3 ligand), Pam3CSK4 (TLR2 ligand), and ODN (TLR9 ligand) were not affected by the presence of TLR5 (*Figure 5B*).

## TLR5 engages with MyD88 after LPS exposure and promotes activation of the MyD88 pathway

We then interrogated the effect of TLR5 on TLR4 signaling. TLR4 signals through both MyD88 and TRIF pathways, whereas TLR5 is so far only known to signal through MyD88 (*Kawai and Akira, 2010*). We found that, in primary murine and human BMDMs, MyD88 co-immunoprecipitated with TLR5 after ultrapure LPS exposure (*Figure 6A*, *Figure 6—figure supplement 1*). This association was directly dependent upon TLR4 activation, since it was not observed in *Tlr4*-deficient primary BMDM (*Figure 6B*). TLR4, as expected, also immunoprecipitated with MyD88 after LPS exposure (*Figure 6B*, *Figure 6—figure supplement 1*). Furthermore, the association of MyD88 with TLR4 and TLR5 was specific, since other TLR like TLR6 and TLR7 did not immunoprecipitate with MyD88 after LPS exposure (*Figure 6B and C*). MyD88 signaling occurs through the complexing of Myd88 with

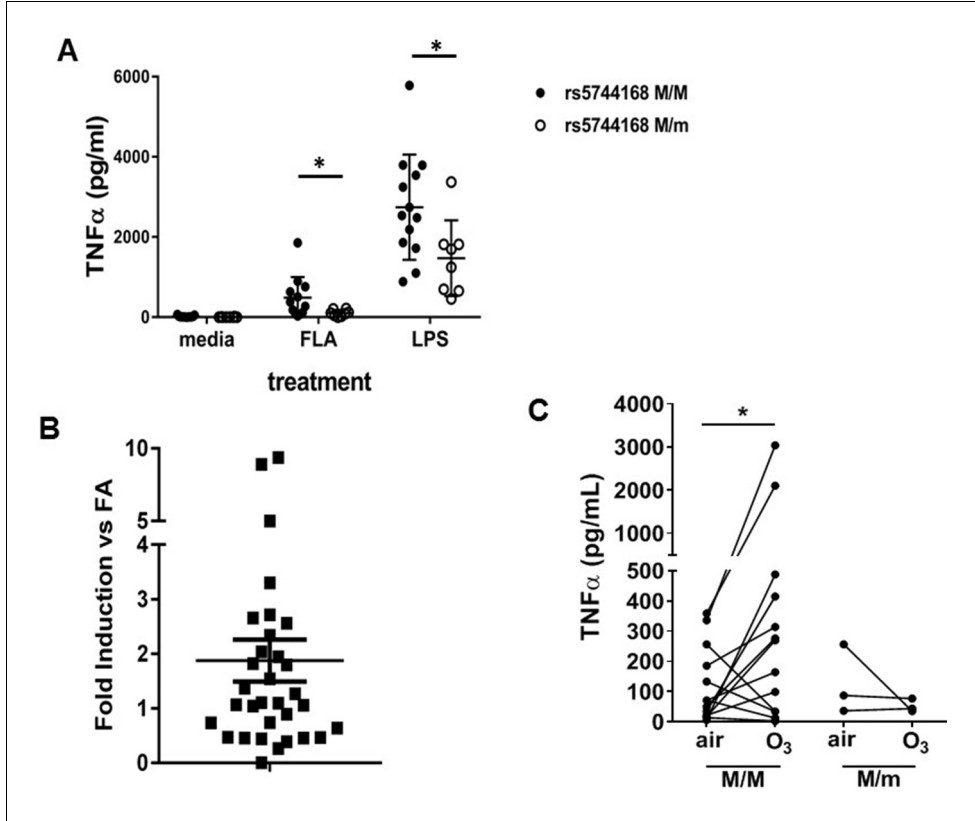

**Figure 4.** TLR5 participates in TLR4-mediated inflammation in humans. (A) TNF-α production by peripheral blood monocyte-derived macrophages from human volunteers either homozygous for the major allele (rs5744168 M/M) or carriers of the minor allele (rs5744168 M/m) for the TLR5 single nucleotide polymorphism rs5744168. Cells were exposed to 10 ng/ml ultrapure LPS or 100 ng/ml ultrapure flagellin for 24 hr and TNF-α levels were analyzed by Duoset ELISA kit. Data are represented as mean ± standard deviation and analyzed by unpaired t test with Welch's correction. N = 7–13 individual subjects. (B) TLR5 gene expression in alveolar macrophages from human volunteers exposed to 200 ppb $O_3$ for 135 min, referenced to TLR5 expression in the same individuals after air exposure. N = 32 individual subjects. Data are presented as individual values with mean ± s.e.m. and was analyzed by Wilcoxon pairwise signed rank test. (C) Ex vivo TNF-α production by human alveolar macrophages after exposure to air or $O_3$ n = 3 minor rs5744168 allele carriers and 20 major allele carriers. Data are represented as individual values and trends and analyzed by Wilcoxon matched-pairs signed rank test. *p<0.05.
The online version of this article includes the following figure supplement(s) for figure 4:

**Figure supplement 1.** TNFa secretion from monocyte-derived macrophages from human subjects depending on TLR5 status.

IRAK-4 which stabilizes formation of the so-called Myddosome (*Gay et al., 2011*). *Tlr5*-deficient BMDM had decreased immunoprecipitation of IRAK-4 with MyD88 (*Figure 6C*). There was also significant reduction in the phosphorylation of IKKα/β, IκB, p65, JNK1/2 and ERK1/2 in *Tlr5*-deficient BMDMs compared with wildtype cells during early activation (*Figure 7A,B*, *Figure 7—figure supplement 1A*), which is MyD88- but not TRIF-dependent (*Kawai et al., 2001*; *Yamamoto et al., 2003*). Nuclear IRF3 (a specific readout of TRIF-dependent signaling *Honda and Taniguchi, 2006*) was not affected in *Tlr5*-deficient cells (*Figure 7—figure supplement 1B*). In aggregate, these results support that TLR5 directly interacts with MyD88 after LPS exposure and enhances MyD88-dependent TLR4 signaling by promoting efficient assembly of Myddosome.

## TLR5 is part of the TLR4 signaling complex

We then investigated whether TLR5 participates directly in the TLR4 signaling complex, or whether it affects TLR4 signaling indirectly. We first evaluated whether TLR5 affects TLR4 cell surface expression and trafficking using bone marrow-derived macrophages (BMDM) from genetically deficient or

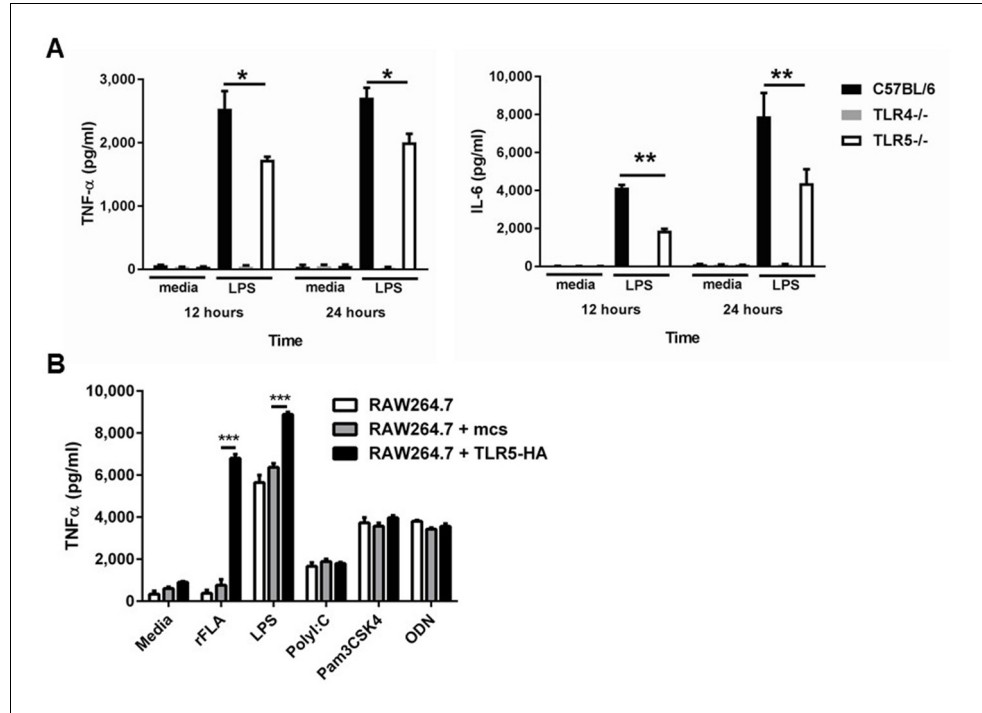

**Figure 5.** Deficiency of TLR5 reduces ultrapure LPS induced inflammatory cytokine production. (A) TNFα and IL-6 production by BMDM from *Tlr5*-deficient (TLR5[-/-]), *Tlr4*-deficient (*Tlr4*[-/-]) and wild-type (C57BL/6) mice after 10 ng/mL ultrapure LPS or vehicle (media) exposure for 12 or 24 hr. N = 8 per group, experiment repeated at least 5 times. (B) TNF-α secretion by non-transfected (RAW264.7), empty construct transfected (RAW264.7 + mcs) or hemagglutinin-tagged TLR5 construct (TLR5-HA) transfected cells (RAW264.7 + TLR5-HA) after vehicle (media) or TLR5, TLR4, TLR3, TLR2 and TLR9 agonists (100 ng/mL recombinant ultrapure flagellin (rFLA), 10 ng/ml LPS, 10 μg/ml PolyI:C, 1 μg/ml Pam3CSK4, 1 μM ODN). N = 8 per group, experiment repeated twice. Data are represented as mean ± s.e.m. and were analyzed by two-way analysis of variance (ANOVA) followed by Tukey's post hoc test. *p<0.05, **p<0.01, ***p<0.001.

The online version of this article includes the following figure supplement(s) for figure 5:

**Figure supplement 1.** TNFa secretion from murine bone marrow-derived macrophages in response to monophoryl lipid A (A) and to flagellin (B).

wild-type mice. There were no differences in basal levels of cell surface TLR4 or CD14 between *Tlr5*-deficient and –sufficient BMDM, nor in LPS-induced internalization of TLR4 (*Figure 8—figure supplement 1A–D*). Because we were unable to find commercially available validated antibodies against TLR4 that could be used in co-immunoprecipitation experiments, we utilized an induced expression system using tagged TLR4 and TLR5 in HEK293 cells and found that TLR4 and TLR5 reciprocally co-immunoprecipitated in transfected HEK293 cells (*Figure 8A*). We then overexpressed TLR4 and TLR5 in HeLa cells and confirmed their interaction through a Proximity Ligation Assay (*Figure 8—figure supplement 1E*). Furthermore, we exposed TLR5-hemagglutinin tagged expressing RAW264.7 cells to biotin-tagged LPS, and (after thoroughly washing the cells) could co-precipitate TLR5 with LPS (*Figure 8B*). These results suggest that TLR5 directly participates in the TLR4 signaling complex after LPS exposure, and, in aggregate, support a functional interaction of TLR5 with TLR4 in the response to environmental injury.

## Discussion

The important novel finding from our work, is that TLR5 heteromerization with TLR4 modulates canonical TLR4 signaling and promotes activation of the MyD88 pathway. Recent evidence highlights the role of molecules of the TLR4 receptor complex in modulating TLR4 signaling. For example, elegant work has demonstrated that CD14, which is necessary for LPS binding to TLR4, also controls

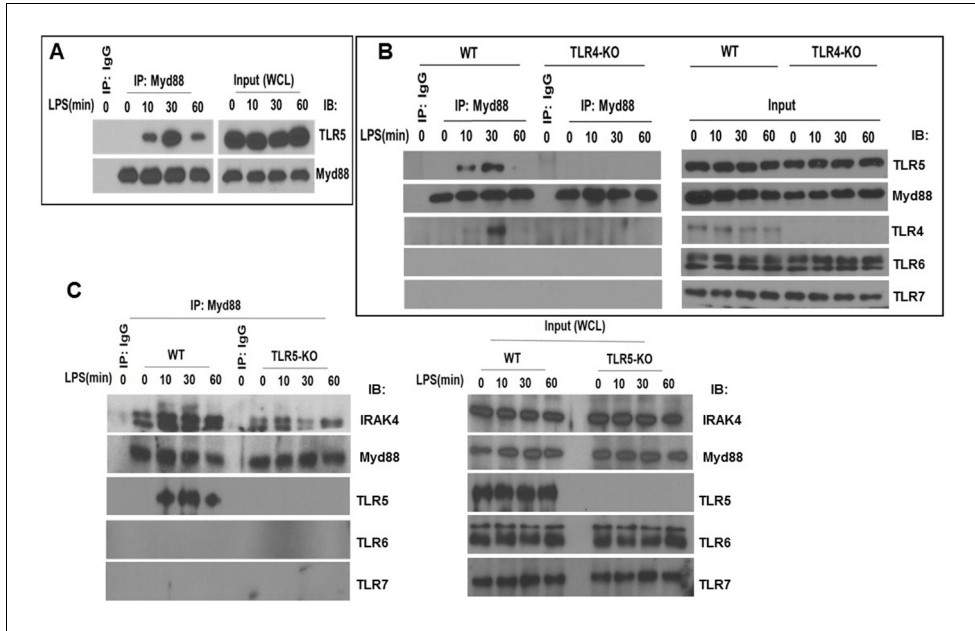

**Figure 6.** TLR5 engages with MyD88 and promotes Myddosome assembly after TLR4 activation. (**A**) Co-immunoprecipitation of TLR5 with Myd88 in BMDM from C57BL/6J mice after 100 ng/mL LPS exposure for indicated time points. n = 7, experiment was repeated twice. (**B**) Immunoprecipitation of TLR5 with Myd88 in BMDM from *Tlr4*-deficient (TLR4-KO) or *Tlr4*-competent (C57BL/6, wildtype WT) mice after 100 ng/mL LPS exposure for indicated time points. Representative of 3 separate experiments. (**C**) Immunoprecipitation of IRAK4 with Myd88 in BMDM from *Tlr5*-deficient (TLR5 KO) or *Tlr5*-competent (C57BL/6, wildtype WT) mice after 100 ng/mL LPS exposure for indicated time points.
The online version of this article includes the following figure supplement(s) for figure 6:

**Figure supplement 1.** TLR4 and TLR5 co-immunoprecipitate with MYD88 after LPS exposure in primary human monocyte-derived macrophages.

TLR4 endocytosis after LPS ligation, and thus is necessary for TRIF signaling, which is thought to occur in the endosomal compartment (*Zanoni et al., 2011*). Our work further suggests that TLR4 signaling is modulated by the addition of TLR5 to the receptor complex. Unlike CD14, TLR5 does not appear to regulate LPS-induced internalization of TLR4 (*Figure 8—figure supplement 1C*). Our findings rather support a model in which TLR5 promotes TLR4/MyD88 signaling at the plasma membrane by enhancing the assembly of the Myddosome. Importantly, our work suggests that TLR5 regulation of TLR4 signaling is biologically significant. TLR5-deficient mice had approx. 30–50% decreased cytokine expression in local and systemic LPS models of lung inflammation, while airway hyperresponsiveness after ozone or hyaluronan exposure was significantly reduced, and inflammatory gene induction after hyaluronan exposure was abolished. It is possible that the 'fine-tuning', MyD88-promoting effects of TLR5 are particularly evident in lower-grade inflammation such as ozone- or hyaluronan-induced, which explains the larger impact on TLR5 deficiency on human and murine inflammation after ozone exposure compared to LPS exposure.

Using primary murine and human macrophages, we demonstrate that, in the physiological state, TLR5 co-immunoprecipitates with MyD88 after ultrapure LPS exposure, but only in the presence of TLR4. This indicates that TLR5 is recruited into the Myddosome assembly, along with TLR4, upon TLR4 activation. Indeed, it has been postulated that the ability of the Myddosome to form 7:4 and 8:4 MyD88:IRAK4 stoichiometries is a potential mechanism through which clusters of activated TLR receptors can be formed and different TLR receptors can be recruited into the same assembly (*Motshwene et al., 2009*). Higher-order assembly of receptor complexes in lipid raft microdomains is likely to be crucial in the fine-regulation of immune responses (*Gay et al., 2011*). Our work suggests that TLR5 may be part of the higher-order receptor assembly that regulates TLR4 signaling. TLR4 is a promiscuous receptor, having been found to heterodimerize with TLR2 and TLR6

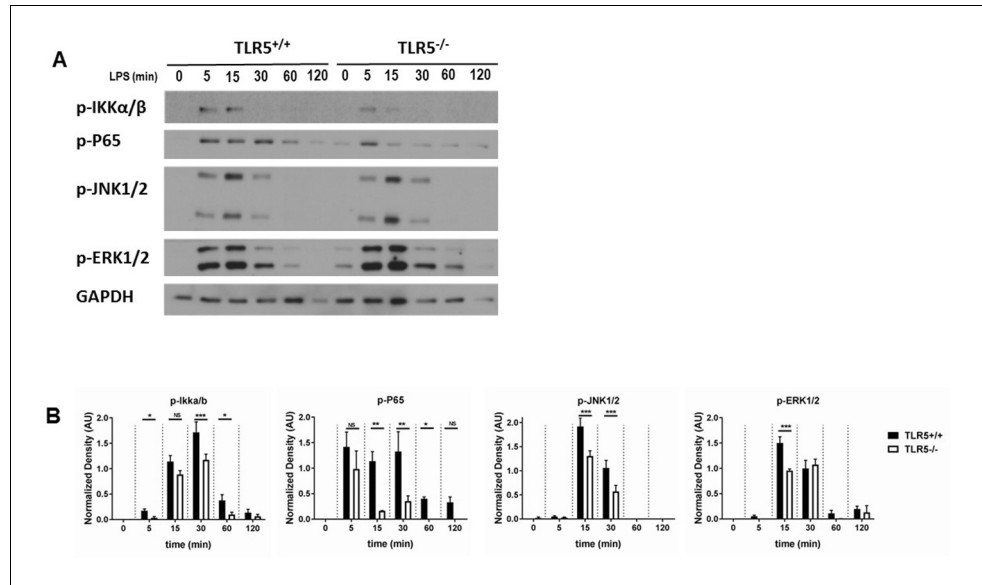

**Figure 7.** TLR5 promotes MyD88 downstream signaling. (**A**) Western blot analysis of p-P65, p-IKKα/β, p-JNK1/2 and p-ERK1/2 after exposure to 100 ng/mL ultrapure LPS exposure in BMDMs from *Tlr5*-competent (TLR5$^{+/+}$) and *Tlr5*-deficient (TLR5$^{-/-}$) mice. (**B**) Quantification of densitometric analysis of 3 separate blots similar to (7A). Data are represented as mean ± s.e.m. and were analyzed by repeated unpaired t test with Holm-Sidak correction. NS = not significant, *$p<0.05$, **$p<0.01$, ***$p<0.001$.

The online version of this article includes the following figure supplement(s) for figure 7:

**Figure supplement 1.** Analysis of downsteam signaling after TLR4 activation, dependent on TLR5 status.

(*Stewart et al., 2010*; *Wang et al., 2014*) but the effect of heteromerization until now has always been to expand the TLR4 ligand spectrum. To our knowledge, this is the first work to demonstrate that TLR heteromerization may serve to modulate canonical TLR signaling. Further research will be necessary to uncover the precise mechanism of this effect. TLR5 heteromerization with TLR4 may help recruit additional MyD88 moieties to the signaling complex, and thus promote downstream signaling. Alternatively, or perhaps in addition, endogenous activators of both TLR4 and TLR5, like

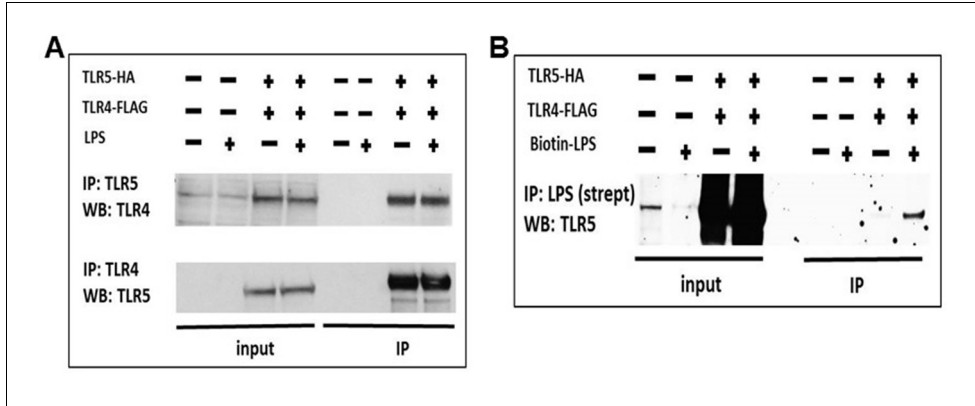

**Figure 8.** TLR5 participates in TLR4 signaling complex. (**A**) Co-immunoprecipitation of hemagglutinin-tagged TLR5 (TLR5-HA) and FLAG-tagged TLR4 (TLR4-FLAG) in HEK293 cells after 100 ng/mL ultrapure LPS exposure. (**B**) Immunoprecipitation of TLR5 with biotinylated ultrapure LPS (Biotin-LPS) in TLR5-HA and TLR4-FLAG transfected HEK293 cells after 100 ng/mL Biotin-LPS exposure for 15 min. Representative of 2 separate experiments.

The online version of this article includes the following figure supplement(s) for figure 8:

**Figure supplement 1.** Cell surface expression of TLR4 and its co-localization with TLR5.

hyaluronan or HMGB1 (*Das et al., 2016*) may engage these receptors and promote the signaling response.

A recent paper supported the role or TLR5 in TLR4 signaling, showing similar effects of the rs5744168 minor allele polymorphism on TNFα and IL-8 expression by human monocytes after LPS exposure, and also demonstrating that TLR5 does not influence the gene expression of TLR4 (45), as we also show. By contrast, that paper could not confirm that TLR5 is modifying the NFκB pathway. This apparent discrepancy may be due to different research methodologies: These authors used gene silencing which only resulted in approx. 50% reduction of TLR5 expression, as well as reporter cell lines with transfection-induced TLR5 expression, as opposed to our use of primary cells with genetically knocked-out gene function. Furthermore, the authors of that study did show an effect for TLR5 on NFκB when transfecting with lower (more physiological) doses of TLR5 DNA (*Dickey et al., 2019*).

In summary, our results suggest a new model of TLR4-TLR5 complex formation in response to the PAMP LPS or the DAMP hyaluronan (*Figure 9*). We propose that the hitherto accepted model of TLR4 signaling through pure TLR4 homodimers rather reflects the TLR5-deficient state. In TLR5-expressing cells, TLR5 participates in a heteromeric higher-order TLR4 receptor complex and potentiates MyD88 signaling by promoting efficient assembly of the Myddosome. This may suggest that exposures that induce TLR5 expression, such as DNA injury, p53 activation (*Menendez et al., 2011*) or flagellated bacterial infection may in parallel prime the MyD88-dependent pro-inflammatory response to LPS, due to the presence of more TLR5 receptors which may promote TLR4 signaling. Since TLR5 signals through MyD88, in a finite TLR4 receptor pool the presence of TLR5/TLR4 higher-order complexes would bias TLR4 signaling towards MyD88. Notably, cell surface expression of TLR4 on immune cells is low (a few hundred or thousand molecules per cell) compared to other TLRs (*Visintin et al., 2001*; *Juarez et al., 2010*), supporting that the presence of relatively few TLR5 receptor molecules may suffice to augment MyD88 signaling downstream of TLR4.

## Materials and methods

### Mice

C57Bl/7J mice and B6.129S1-*Tlr5*^tm1Flv^/J (TLR5-deficient) mice were purchased from the Jackson Laboratory (Bar Harbor, ME). The TLR5-deficient allele was generated in the 129S1 genome and subsequently backcrossed to C57BL/6 before being transferred to the Jackson Laboratory by the donating investigator. When possible, wildtype littermate control mice were used in our study in parallel to commercial wildtype C57BL/6J controls. To ensure that our results were not due to locality-influenced microbiome changes, we repeated experiments in mice that were bred at the NIEHS vivarium, as well as mice purchased from the Jackson Laboratory and studied within 1 week of arrival. In some experiments, mice received neomycin water to control gut microbiome. Results were comparable independent of provenance or antibiotic dosing. Mice were given access to water and chow ad libitum and were maintained at a 12 hr dark-light cycle. No differences in body weight were observed at the ages studied (6–12 weeks old). All experiments are approved by the NIEHS Institutional Animal Care and Use Committee.

### Exposures

Mice received ultrapure *E. coli* O111:B4 LPS (List Biological Labs, Campbell, CA) (50 µl of 1 mg/ml in PBS), or control PBS vehicle only, by oropharyngeal aspiration and were phenotyped 24 hr later. For systemic LPS exposure, mice received 10 mg/kg LPS or PBS by intraperitoneal injection and were phenotyped 24 hr later. In other experiments, mice were exposed to 2 ppm ozone for 3 hr, in a chamber with 20 exchanges/hour, 50–65% relative humidity and a temperature of 20–25° C as previously described (*Garantziotis et al., 2009*) and were phenotyped 24 hr later. Control mice received filtered air in an identical setup. In other experiments, mice received 50 µl of a 3 mg/ml solution of sonicated, LPS-free, pharmaceutical-grade hyaluronan with an average molecular weight of 100–300 kDa (derived from Healon, Abbott Laboratories, Abbott Park, IL) or PBS vehicle by retropharyngeal aspiration, and were phenotyped 2 hr later.

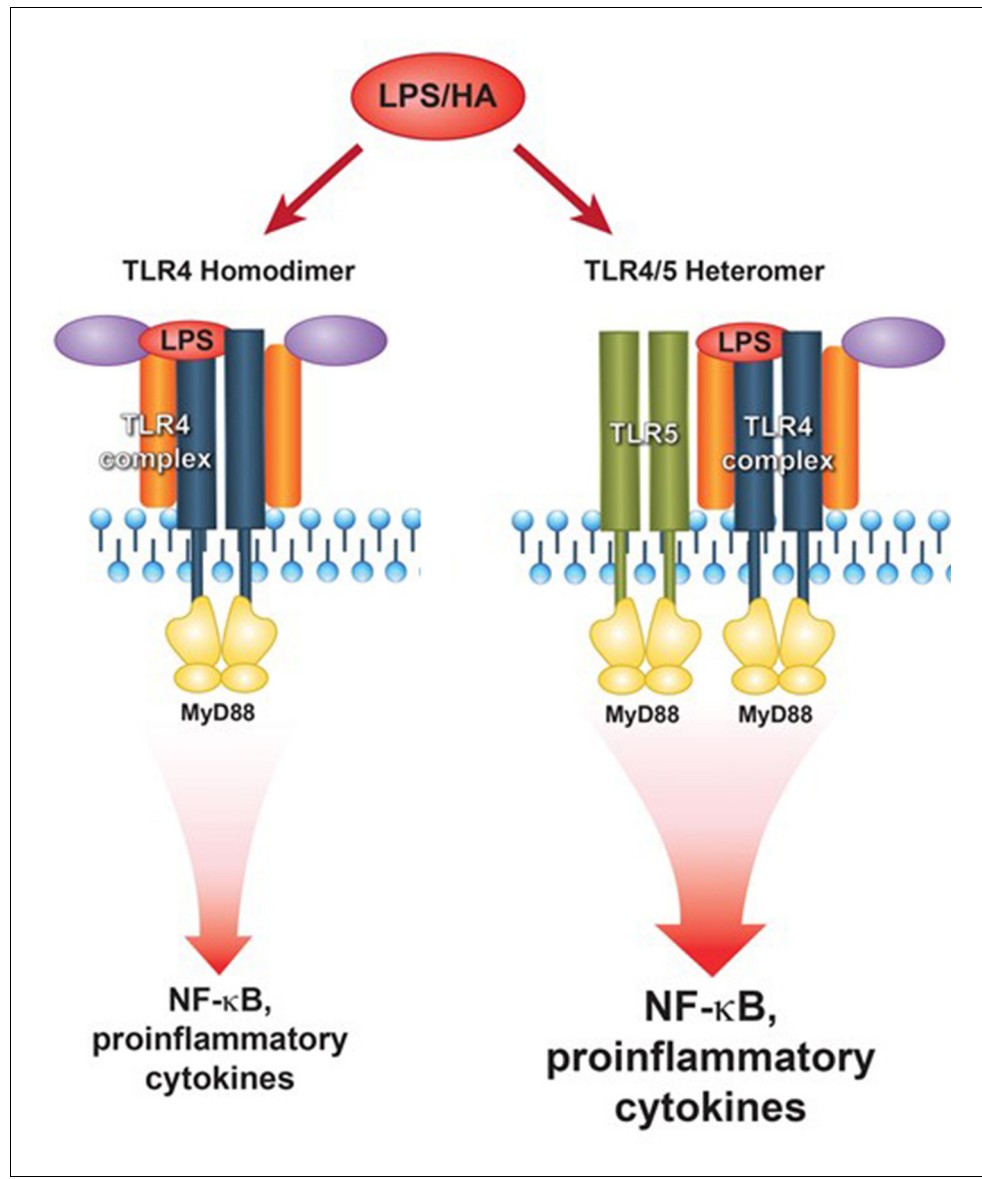

**Figure 9.** Proposed model of TLR5-TLR4 interaction during environmental lung injury. The current model of canonical TLR4 activation rather applies to the TLR5 deficient status (left panel). In the presence of TLR5 (right panel), TLR5 participates in the TLR4 signaling complex, and promotes signaling downstream of the MyD88 pathway.

## Airway physiology measurements

Airway responsiveness to 25–100 mg/ml methacholine (MCh; Sigma) was measured 24 hr following $O_3$ or 2 hr following hyaluronan challenge. Briefly, mice were anesthetized with urethane (2 g/kg; Sigma), tracheotomized with a tracheal cannula (Harvard Apparatus) with Luer adapter, and mechanically ventilated on a 42°C water-heated pad at a rate of 150 breaths/min, a tidal volume of 10 ml/kg and a positive end-expiratory pressure (PEEP) of 3 cm $H_2O$ with a computer-controlled small animal ventilator (FlexiVent, Scireq, Montreal, Canada). To block spontaneous breathing, mice were given pancuronium bromide i.p. (0.8 mg/kg; Sigma-Aldrich) 5 min prior to assessment of airway responses. To measure airway responsiveness, a single-frequency forced oscillation waveform, followed by a broadband forced oscillation waveform (matched to the animal breathing frequency) were applied using the Flexiware 7.6 software default mouse inhaled dose-response script. The resulting pressure, volume, and flow signals were fit to either the Single Compartment or Constant Phase model of the

lung to obtain total respiratory system resistance ($R_{rs}$) and elastance ($E_{rs}$) or Newtonian resistance ($R_n$, generally understood as proximal airway resistance), tissue damping (G, generally understood as peripheral tissue resistance), and tissue elastance (H), respectively (*Irvin and Bates, 2003*). The peak response at each dose was averaged and graphed along with the average baseline measurement for each group.

## Proximity ligation assays (PLA, Duolink)

TLR4-FLAG M2 or MyD88-V5 and TLR5-HA were overexpressed in HeLa cells grown on glass coverslips. 24 hr post-transfection the cells were fixed with 4% paraformaldehyde (PFA) for 10 min at room temperature (RT) and blocked with 10% normal goat serum for 1 hr at RT. The cells were next permeabilized with 0.1% TritonX-100 in goat serum for 15 min at RT and incubated with primary antibodies (dilution 1:1000) against epitope-tags overnight: rabbit anti-FLAG M2 (Cell Signaling), mouse anti-V5 (Invitrogen) and mouse anti-HA (Sigma). Duolink, based on in situ proximity ligation assay (PLA), was performed according to manufacturer instructions (Sigma).

## Bone Marrow Derived Macrophage (BMDM)

Bone marrow was collected from tibias and femurs of wildtype *and Tlr5*-deficient mice and cell single preparations were made. Growth medium for maturation of BMDM consisted of DMEM-F12 containing 10 mM L-glutamine, 10% embryonic stem cell qualified fetal bovine serum, 1% antibiotic and antifungal mix and 30 ng/mL murine M-CSF. Cells were cultivated in an incubator at 37°C, 5% $CO_2$ for up to 7 days with medium change every 48 hr after first medium change 72 hr after platting. *Tlr5*-deficient BMDM were evaluated for responsiveness to flagellin and were found to be unresponsive (*Figure 5—figure supplement 1A*).

## Use of cell lines

We used HeLa cells and Raw264.7 cells. Neither cell line is among the list of commonly misidentified cell lines. All cell lines were procured by ATCC and were free of mycoplasma contamination.

## Flow cytometry

In order to access cell surface expression of TLR4, Wildtype, *Tlr4*-deficient or *Tlr5*-deficient BMDM were harvested, washed with PBS and were exposed to ultrapure LPS for 0,15,30, 60 or 90 min. Cells were washed with cold PBS and gently lifted from the culture dishes using a cell lifter. Cells numbers were estimated and cells were aliquoted in $1 \times 10^6$ cells per tube in the FACS buffer (0.5% BSA, 0.1% NaN3, and 2 mM EDTA in PBS). Cells were blocked for 20 min on ice in a blocking solution (FACS buffer, 10% species specific serum, and 1% FCR block). Cells were stained using APC anti-mouse CD284 (TLR4) Antibody (clone SA15-21), anti-CD14 Antibody (Biolegend) or isotype controls for 30 min on ice. Cells were washed two times with 1 mL FACS buffer after staining, suspended in 500 uL FACS buffer containing 1 mg/mL propidium iodide (to identify dead cells) and analyzed on a BD FACSAria II equipment.

## Western blotting

Wildtype or *Tlr5*-deficient BMDM were harvested, washed once with cold PBS, and lysed for 30 min at 4°C in 1% TritonX-100, 20 mM HEPES (pH 7.4), 150 mM NaCl, 1.5 mM $MgCl_2$, 2 mM EGTA, protease and phosphatase inhibitors (Roche). Cellular debris was removed by centrifugation at 16,000rcf for 10 min. For immunoblotting, cell extracts were fractionated by SDS-PAGE and transferred to Immobilon-P transfer membranes (Millipore), using either a wet transfer apparatus (Bio-Rad) or with a dry transfer system (iBlot) from Invitrogen. Immunoblot analysis was performed, and the bands were visualized with HRP-coupled goat anti-rabbit, goat anti mouse, or donkey anti-goat Ig as appropriate (Rockland), using the ECL western blotting detection system (GE Healthcare). Protein levels were equilibrated with the Protein Assay Reagent (Bio-Rad).

## Co-immunoprecipitation

For coimmunoprecipitations, cells were harvested, washed once with cold PBS, and lysed in a TritonX-100-containing buffer (0.5% TritonX-100, 20 mM HEPES (pH 7.4), 150 mM NaCl, 1.5 mM $MgCl_2$, 2 mM EGTA, protease and phosphatase inhibitors (Roche). Cell extracts were incubated with

1 µg of Ab (anti-HA, Sigma) or normal IgG (negative control) for 2 hr, followed by incubation for 12 hr with 30 µl of protein G-Sepharose beads (prewashed and resuspended in lysis buffer at a 1:1 ratio). After incubations, the beads were washed four times with lysis buffer, separated by SDS-PAGE, and analyzed by immunoblotting. For TLR4, TLR5, TLR6 and TLR7 antibodies, blocking agent was 5% BSA and antibody dilution was 1:10,000. For Myd88 antibody, blocking agent was 5% milk and antibody dilution was 1:1000. Antibodies used were TLR4: Cat#482300 (Life Technology). TLR5: Cat#PA1-41139 (Invitrogen), TLR6: Cat#AF1533 (R and D Systems), TLR7: Cat# MAB7156 (R and D Systems), Myd88: Cat#4283 (cell signaling).

## ELISA assays

ELISA assay was performed using either R and D Duoset assay kits or Luminex multiplex assays according to manufacturer recommendations.

## Human ozone exposure, Alveolar Macrophage Isolation, Culture

The complete details of the human exposure studies and the subject characteristics were previously published (*Frush et al., 2016*). After obtaining informed consent through a Duke University Institutional Review Board approved protocol, healthy human subjects were exposed to filtered air and ozone (200 parts per billion) in a crossover challenge designed study. Exposures were for 135 min, during which participants alternated between resting and walking on a treadmill at 2–3 mph to mimic an individual performing mildly strenuous activity under ambient conditions. Ozone was created from a 100% O2 source by cold plasma corona discharge (Ozotech, Yreka CA), and mixed with filtered air before addition to chamber and was continuously monitored. The order of filtered air or ozone exposure was randomized for every participant, with at least a 21 day washout period. Approximately 20 hr after exposure, participants underwent a flexible bronchoscopy with bronchoalveolar lavage. Following bronchial alveolar lavage, human alveolar macrophages were isolated. After red cell lysis and counting, the macrophages were re-suspended in medium (RPMI1640 with 10% heat-inactivated FBS, 100 units/ml penicillin, and 100 µg/ml streptomycin) and plated in a 24 well plate at a density of 200,000 cells per well. The cells were maintained in a CO2 incubator at 37° C for 2 hr. After 2 hr, the medium was replaced to remove non-adherent cells and then 2 hr later, the supernatant was collected and the cells were harvested for RNA. RNA extraction was performed using the Fourth Edition Qiagen Protocol (Qiagen, RNeasy Mini Kit, 4th edition, Valencia, CA), followed by DNase treatment (DNase I, Ambion, Austin, TX) and cDNA synthesis (BioRad). RT-PCR was performed on an ABI SDS 7500 (Applied Biosystems) using SYBR Green Reagent (Clontec Laboratories Inc, Mountain View, CA). TLR5 expression was determined in comparison to the 18 s RNA housekeeping gene and the data reported as fold change over the matched filter air sample for each individual subject. The following primers were used for RT-PCR: 18 s (Fwd: GTAACCCG TTGAACCCCATT, Rev: CCATCCAATCGGTAGTAGCG); TLR4 (Fwd: GGCCATTGCTGCCAACAT, Rev: CAACAATCA CCTTTCGGCTTTT), TLR5 (Fwd: TGTATGCACTGTCACTCTGACTCTGT, Rev: AGCCCCGGAACTTTGTGACT). Human TNF-α was measured from the cell supernatants via ELISA (MAX Standard Set kit, BioLegend) according to manufactures instructions. Readings were taken using BMG LABTECH Omega (Software Version 1.20). In a second study, participants were invited according to their rs5744168 genotype. Peripheral blood was drawn, and monocyte-derived macrophages isolated after 7 days in culture and exposed to ultrapure LPS or flagellin as indicated.

## Statistical analyses

Data are represented as mean ± s.e.m. and were analyzed depending on experimental design by either analysis of variance (one-way or two-way ANOVA) followed by Tukey's post hoc test or by unpaired t test with Welch's or Holm-Sidak correction as appropriate. Tlr5 gene expression and TNF-α production data from ozone expose human volunteer macrophages is presented as individual values and analyzed by Wilcoxon pairwise signed rank test.

## Study approval

All clinical studies described in this work were approved by the Institutional Review Boards of the NIEHS and Duke University respectively. Written informed consent was received from all participants prior to inclusion in the described studies.

## Acknowledgements

Authors thanks Kevin Katen, James Ward, Rickie Fannin and Ligon Perrow for their excellent technical assistance.

## Additional information

### Funding

| Funder | Grant reference number | Author |
|---|---|---|
| National Institute of Environmental Health Sciences | Z01ES102605 | Stavros Garantziotis |
| National Institute of Environmental Health Sciences | Z01ES102005 | Michael B Fessler |

The funders had no role in study design, data collection and interpretation, or the decision to submit the work for publication.

### Author contributions

Salik Hussain, Collin G Johnson, Joseph Sciurba, Xianglin Meng, Vandy P Stober, Caini Liu, Jaime M Cyphert-Daly, Katarzyna Bulek, Wen Qian, Alma Solis, Yosuke Sakamachi, Carol S Trempus, Jim J Aloor, Kym M Gowdy, Michael B Fessler, Formal analysis, Investigation; W Michael Foster, John W Hollingsworth, Xiaoxia Li, Funding acquisition, Investigation, Project administration; Robert M Tighe, Formal analysis, Funding acquisition, Investigation, Project administration; Stavros Garantziotis, Conceptualization, Supervision, Funding acquisition, Investigation, Methodology, Project administration

### Author ORCIDs

Katarzyna Bulek (iD) http://orcid.org/0000-0001-8064-7047
Xiaoxia Li (iD) http://orcid.org/0000-0002-4872-9525
Stavros Garantziotis (iD) https://orcid.org/0000-0003-4007-375X

### Ethics

Clinical trial registration NCT01087307, NCT00341237, NCT00574158.

Human subjects: All subjects signed informed consent and all clinical research protocols were approved by the IRBs at Duke University Medical Center and the National Institute of Environmental Health Sciences, as applicable. The study described herein is using data collected as part of several clinical or translational studies (NCT01087307, NCT00341237, NCT00574158) and were approved by NIEHS and Duke IRBs (Protocol IRB approvals # 10-E-0063, 04-E-0053, 12496-CP-004).

Animal experimentation: Mice were given access to water and chow ad libitum, and were maintained at a 12-hour dark-light cycle. All experiments are approved by the NIEHS Institutional Animal Care and Use Committee.

### Decision letter and Author response

Decision letter https://doi.org/10.7554/eLife.50458.sa1
Author response https://doi.org/10.7554/eLife.50458.sa2

## Additional files

### Supplementary files

• Supplementary file 1. List of genes.

• Transparent reporting form

### Data availability

Source data files have been provided.

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
