## [Decision Letter]

**Acceptance summary:**

Your revision now provides sound evidence for the involvement of TLR5 in the TLR4 signaling. Your additional experiments have taken away the concerns that the TLR5 participation found was due to artifacts like effects of contaminating TLR5 ligands.

The concept that TLR5 is not just a flagellin receptor, but has a sizable role in MYD88 dependent TLR4 signaling is of importance for pathology, such as lung injury by substances like ozone. Based on this work, it is to be expected that a role for TLR5 in other pathology will be pursued by others as well.

**Decision letter after peer review:**

Thank you for submitting your article "TLR5 participates in the TLR4 receptor complex and biases towards MyD88-dependent signaling in environmental lung injury" for consideration by *eLife*. Your article has been reviewed by Tadatsugu Taniguchi as the Senior Editor, a Reviewing Editor, and two reviewers. The reviewers have opted to remain anonymous.

The reviewers have discussed the reviews with one another and the Reviewing Editor has drafted this decision to help you prepare a revised submission.

In this paper the function of TLR5 as part of the TLR4 signaling pathway for LPS and for hyaluronan recognition is described. Using TLR5 KO mice and human subjects carrying a TLR5 mutation LPS signaling was found to be inhibited, in vitro as well in vivo. Data are provided that TLR4/TLR5 complex could be formed after stimulations.

Essential revisions:

1) The central issue with the paper is whether the 'ultrapure' LPS is free from contamination for instance with flagellin, DNA and PGN. The LPS purity issue was addressed in vitro with the TLR4 KO control, but the reviewers consider that in-vivo proof is also necessary. A TLR 4-blocking experiment in TLR5-/- mice would be needed to see whether there is a pure TLR4 response. Likewise testing LPS from other sources is essential.

2) A second major issue is the ability to demonstrate that there is real TLR4/5 synergy in vitro and in vivo. The BMDM studies in vitro argue against a synergy model, since it is not likely that DAMPs that could be synergizing with LPS are being released in a short time frame.

3) A third point is the question what triggers TLR5 in the models used if there would be no flagellin moieties around? Endogenous ligands of TLR5 that have been described in the literature should be considered (and be mentioned in the introduction and the discussion. Could hyaluronan have direct or indirect TLR5 activating properties?

4) The reviewers feel that there are some invalid assumptions in the interpretation of the downstream signaling results regarding MyD88 and NFkB. The conclusion that TLR5 is driving TLR4-MyD88 but not TLR4-TRIF dependent signaling, may be an overstatement. Only data are provided that the TRIF-dependent arm is not affected is the lack of a change in IRF3 localization. This does not preclude an effect on TRIF-NFkB signaling. Some (potentially time-consuming) experiments that would really clarify this point would be to determine if TLR5 can bind to TRIF and examine the phenotype of BMDM from a TLR5/MyD88 double mutant mouse. These experiments are not strictly necessary if the interpretation/conclusions are appropriately tempered.

5) Along those same lines, the TRIF pathway can affect NFkB as well as IFN signaling, so the authors should rephrase how they deal with this point in subsection “TLR5 promotes TLR4-mediated inflammation and airway hyperresponsiveness in vivo”, subsection “TLR5 modulates TLR4-dependent signaling.”, Discussion section and in their model Figure 9.

6) Somewhat related is a caveat about the nanostring interpretation. Because NFkB-dependent genes on the nanostring data are over-represented, the authors conclude that NFkB is being regulated directly. Alternatively, since these data are from whole lung tissue, this could just be reflective of decreased recruitment of inflammatory cells.

7) In Figure 1, there is no protein levels shown of pro inflammatory mediators only mRNA expression, why? In the O_3_ experiments (Figure 2), histology of the inflamed lungs is missing.

8) The co-localization of TLR4 and TLR5 was demonstrated by using overexpression system. The authors should show that TLR4/5 complexes are present in primary human cells.

9) In the figure legends it should be mentioned how many mice per group were used and how many times the experiment was repeated. E.g., Figure 1, N=5-8 mice per group, experiment repeated twice. Data are represented as mean (plus minus SEM) and were analyzed by unpaired t test with Welch's correction. Does this mean that 15-24 mice were used for Figure 1B and C?

---

## [Author Response]

Essential revisions:1) The central issue with the paper is whether the 'ultrapure' LPS is free from contamination for instance with flagellin, DNA and PGN. The LPS purity issue was addressed in vitro with the TLR4 KO control, but the reviewers consider that in-vivo proof is also necessary. A TLR 4-blocking experiment in TLR5-/- mice would be needed to see whether there is a pure TLR4 response. Likewise testing LPS from other sources is essential.

We provide several lines of evidence to ensure that the results are not due to contamination. First, we would like to note that, besides LPS, we have used two other in vivo exposure models, ozone and hyaluronan. The former is a sterile gas, making contamination impossible, and the latter is a pharmaceutical-grade compound, used for inhalation in human patients, and is therefore also pharmaceutical-grade and thus essentially contaminant-free. Nevertheless, we provide further evidence of absence of contamination. We show the response of TLR4 KO, TLR5 KO and TLR4/5 DKO mice, by lung lavage cytokines, to a different ultrapure LPS, from Invivogen (the previously used one was by List). This information is new Figure 1D. This strongly supports that this is not an independent TLR5 activation effect due to contamination. We also show in new Figure 3C that hyaluronan activation of airway constriction is dependent on TLR4 and cannot occur in the presence of TLR5 alone. Thus, our data support that TLR5 contaminants are not involved in the observed effects.

2) A second major issue is the ability to demonstrate that there is real TLR4/5 synergy in vitro and in vivo. The BMDM studies in vitro argue against a synergy model, since it is not likely that DAMPs that could be synergizing with LPS are being released in a short time frame.

Thank you for the opportunity to clarify. We are not arguing that TLR5 is activated by endogenous ligands and synergizing with LPS-activated TLR4. Rather, we are arguing that TLR5 is participating in the LPS-response complex and promotes TLR4 response to its own ligands. We do not believe that endogenous TLR5 ligands are involved (please also see response to point#3). This is supported by the fact that TLR4 deficiency leads to a complete abolition of the response, thus the observed TLR5 effects are predicated on a functional TLR4.

Editor’s reply regarding point #2:

“Perhaps the term synergy should be avoided. It would probably suffice if you would clarify better how you think TLR5 interacts. No activation of TLR5? No DAMPS involvement?”

Thank you for the opportunity to clarify. Indeed, we believe that our in vitro experiments conclusively show that activation of TLR5 by a PAMP is not involved, since we use by definition sterile systems and ultrapure reagents (LPS or hyaluronan). The issue of a TLR5 activation by a DAMP is not conclusively addressed in this paper, in that it is difficult to prove a negative. It is certainly possible that a DAMP is involved. However, the crux of our paper is that TLR5 participates in the TLR4 signaling complex. Therefore, we believe that the most relevant question here is “does TLR5 get activated by a DAMP that is not also activating TLR4?”. The only 2 DAMPs that have been reported to need TLR5 for signaling are hyaluronan (our paper, first time report) and HMGB1 (Das et al., 2016, see also response below). Both hyaluronan and HMGB1 are TLR4 activators. Therefore, even with a DAMP involvement, a receptor complex of TLR4 and TLR5 is invoked, since they both co-immunoprecipitate with MyD88. We have added these arguments in our discussion.

3) A third point is the question what triggers TLR5 in the models used if there would be no flagellin moieties around? Endogenous ligands of TLR5 that have been described in the literature should be considered (and be mentioned in the introduction and the discussion. Could hyaluronan have direct or indirect TLR5 activating properties?

Editor’s suggestion in response to our query regarding point #3:

“The reviewers are aware of these doubts regarding TLR5. Blocking HMGB1 would be great. Also, a Hyaluronan experiment would help convince the reviewers.”

Flagellin moieties cannot be entirely excluded in vivo. However, we provide several in vitro experiments, both with human and mouse primary cells as well as cell lines, which support the involvement of TLR5 in TLR4 signaling. We do not see a possibility of flagellin contamination in all these experiments. To further enhance this finding, we have performed in vitro tracheal ring experiments, exposing sterile tracheal rings to pharmaceutical-grade hyaluronan (i.e. contaminant-free), which is a TLR4 signaling activator. We show that TLR4 KO and TLR5 KO tracheal rings are as unresponsive to hyaluronan as MyD88 KO tracheal rings. This supports that TLR4 and TLR5 have non-redundant roles in hyaluronan signaling, and in our opinion argues against endogenous TLR5 ligands promoting activation without need for TLR4 activation. Thus, our results suggest that hyaluronan cannot activate TLR5 independent of TLR4.

Specifically, with regard to TLR5 ligands, we were only able to find very few papers in the literature referencing endogenous TLR5 ligands. Most of these papers are from the rheumatoid arthritis literature and are not mentioning specific ligands, but rather inferring the presence of endogenous ligands because TLR5 activity was found in synovial fluid that is supposedly sterile (Kim et al., 2014 and Chamberlain et al., 2012). Please note, that the supposed sterility of synovial fluid is no longer certain (for example Zhao et al., 2018). One paper (Das et al., 2016) mentions HMGB1 as a possible TLR5 agonist. However, please note that HMGB1 is also a TLR4 DAMP (Park et al., 2004 and also the Das et al., paper).

We did perform a HMGB1 inhibition assay in our tracheal ring setup, as suggested by the editor (See Author response image 1). Briefly, we incubated tracheal rings from C57Bl/6J mice with hyaluronan (which sterilely induces tracheal ring constriction via a TLR4/MYD88 pathway) and found that HMGB1 neutralizing antibody inhibits constriction after hyaluronan exposure. However, since HMGB1 is also a TLR4 activator, as mentioned above, I am not sure whether this result is relevant with regard to endogenous TLR5 activation. In other words, this does not prove that HMGB1 is a specific TLR5 endogenous agonist. It just suggests that it mediates hyaluronan/TLR4/TLR5 signaling. Therefore, this experiment does not disprove that TLR5 participates in the TLR4 receptor complex.

**Author response image 1. respfig1:** Neutralization of HMGB1 inhibits hyaluronan-induced airway hyperresponsiveness. Murine tracheal rings were exposed to short-fragment hyaluronan which is known to elicit hyperresponsiveness of smooth muscle to methacholine. The rings were incubated with increasing doses of methacholine (an airway constrictor) and the muscle force generation was assayed, normalized to the standard response to potassium chloride (KCl), a nonspecific constrictor of the muscle. Treatment with the HMGB1-neutralizing antibody reduced the responsiveness to methacholine after hyaluronan exposure to baseline (PBS exposure) levels.

4) The reviewers feel that there are some invalid assumptions in the interpretation of the downstream signaling results regarding MyD88 and NFkB. The conclusion that TLR5 is driving TLR4-MyD88 but not TLR4-TRIF dependent signaling, may be an overstatement. Only data are provided that the TRIF-dependent arm is not affected is the lack of a change in IRF3 localization. This does not preclude an effect on TRIF-NFkB signaling. Some (potentially time-consuming) experiments that would really clarify this point would be to determine if TLR5 can bind to TRIF and examine the phenotype of BMDM from a TLR5/MyD88 double mutant mouse. These experiments are not strictly necessary if the interpretation/conclusions are appropriately tempered.

Thank you for this helpful direction. We have amended our assertions, please see amended title of our paper, amended Figure 9, and new Discussion section. We have removed references to TRIF.

5) Along those same lines, the TRIF pathway can affect NFkB as well as IFN signaling, so the authors should rephrase how they deal with this point in subsection “TLR5 promotes TLR4-mediated inflammation and airway hyperresponsiveness in vivo.”, subsection “TLR5 modulates TLR4-dependent signaling”, Discussion section and in their model Figure 9.

Thank you for this helpful direction. We have amended our assertions, as suggested.

6) Somewhat related is a caveat about the nanostring interpretation. Because NFkB-dependent genes on the nanostring data are over-represented, the authors conclude that NFkB is being regulated directly. Alternatively, since these data are from whole lung tissue, this could just be reflective of decreased recruitment of inflammatory cells.

We have amended the results and discussion to temper our conclusions and enter these caveats.

7) In Figure 1, there is no protein levels shown of pro inflammatory mediators only mRNA expression, why? In the O_3_ experiments (Figure 2), histology of the inflamed lungs is missing.

We have now added the cytokine data, new Figure 1D. Ozone, at the doses we used, leads to very limited or no histological changes, thus including histology would not add any information.

8) The co-localization of TLR4 and TLR5 was demonstrated by using overexpression system. The authors should show that TLR4/5 complexes are present in primary human cells.

Unfortunately, we have been unable to perform IP of TLR4 or TLR5 with existing antibodies, either human or mouse. This is a well-known problem in the TLR4 literature. We discussed this issue with the Editor, and we agreed on an alternative approach: we used human monocyte-derived macrophages and performed immunoprecipitation for MyD88, and western for TLR4 and TLR5. We demonstrate that in primary human cells (monocyte-derived macrophages), TLR4 AND TLR5 co-immunoprecipitate with Myd88 after LPS exposure. Although this is an indirect proof, we feel that this strongly suggests that TLR4, TLR5 and MYD88 are part of the same complex after LPS exposure.

9) In the figure legends it should be mentioned how many mice per group were used and how many times the experiment was repeated. E.g., Figure 1, N=5-8 mice per group, experiment repeated twice. Data are represented as mean (plus minus SEM) and were analyzed by unpaired t test with Welch's correction. Does this mean that 15-24 mice were used for Figure 1B and C?

We apologize for the unclear statements. The analyses were performed for each separate experiment. So, the N depicted was 5-8, as indicated in Figure 1, for example, but in total we used 15-20 mice because we repeated the experiment twice, with similar results.